# Bioconsolidation of Damaged Construction Calcarenites and Evaluation of the Improvement in Their Petrophysical and Mechanical Properties

**DOI:** 10.3390/ma16176043

**Published:** 2023-09-02

**Authors:** Yolanda Spairani-Berrio, J. Antonio Huesca-Tortosa, Carlos Rodriguez-Navarro, María Teresa Gonzalez-Muñoz, Fadwa Jroundi

**Affiliations:** 1Department of Architectural Constructions, University of Alicante, Carretera de San Vicente del Raspeig, s/n, 03690 San Vicente del Raspeig, Spain; ja.huesca@ua.es; 2Department of Mineralogy and Petrology, Faculty of Science, University of Granada, Avda. Fuentenueva s/n, 18071 Granada, Spain; carlosrn@ugr.es; 3Department of Microbiology, Faculty of Science, University of Granada, Avda. Fuentenueva s/n, 18071 Granada, Spain; mgonzale@ugr.es (M.T.G.-M.); fadwa@ugr.es (F.J.)

**Keywords:** bioconsolidation, bacterial carbonatogenesis, stone consolidation, limestone treatment, calcarenite treatment

## Abstract

Bioconsolidation treatment using bacterial carbonatogenesis has been proposed as an environmentally friendly strategy for the efficient preservation of damaged stones, particularly suitable for carbonate stones. The study presented here deals with the evaluation of the performance of this treatment, applied to damaged carbonate stones in two historical buildings in Spain. The methodology applied in this research serves as a reference for future similar studies. Results showed significant improvement in the petrophysical and mechanical properties of the damaged stone following the treatment through the production of calcite and vaterite by the abundant carbonatogenic bacteria inhabiting the stone. These bacteria were able to effectively consolidate weathered areas if an adequate nutritional solution was employed, thereby augmenting the stone’s resistance, as evidenced by the Drilling Resistance Measurement System (DRMS). FESEM images showed calcified bacteria and calcified exopolymeric substances (EPS) consolidating stone minerals without blocking their pores. In addition to consolidation, this biotreatment improves the stone’s behavior against water absorption and increases the contact angle of water droplets without significant modifications in the pore size or diminishing vapor permeability. No color changes are observed. Overall, these results show that the application of the nutritional solution (M-3P) for in situ consolidation of different types of porous carbonate building stones is a highly effective conservation method, with no modification of the chemical composition of the treated materials.

## 1. Introduction

Historical buildings made of calcarenite are particularly susceptible to deterioration, primarily resulting from their interaction with atmospheric agents (e.g., CO_2_ through acidification), pollution (e.g., SO_2_, NO_x_), and salt crystallization, especially in humid environments [1,2]. Numerous studies have demonstrated that this deterioration intensifies significantly over time [1,2,3,4,5,6] due to the stone’s intrinsic properties, such as its weak mineralogical composition and its type of porosity [6,7]. Typically, these characteristics of calcarenite make it particularly susceptible to the action of weathering agents, significantly impacting the appearance of the exposed stone surface. One of the most important factors that cause building stone decay is moisture and rain penetration [8,9]. Over the course of time, spanning decadal to centennial scales, these agents cause a gradual increase in material porosity, which in turn leads to the decohesion and disintegration of mineral constituents. Ultimately, this continuous deterioration process culminates in the destruction of the stone, posing a considerable threat to the structural integrity and historical value of the affected buildings [10]. Restoring buildings made of this type of stone typically involves the common and convenient application of protective coatings and/or conventional consolidants [11,12,13,14,15,16,17]. These treatments aim to enhance the stone’s cohesion, restore surface resistance (lost by the weathering agents), and limit water absorption while preserving its capacity for water vapor permeability [18]. Consequently, over the years, various commercial products have been extensively employed to improve surface properties by creating compounds with different mineralogical compositions. These products include potassium aluminate, sodium and potassium silicate, magnesium, zinc, or aluminum fluosilicate, and several others. However, so far, none of them have been shown to be efficient, and most of them have been observed to produce detrimental side effects on the stone. These side effects encompass a range of issues, including altered structural integrity, compromised surface aesthetics, and accelerated degradation. The interactions of these compounds with the stone’s composition can lead to phenomena such as increased porosity, diminished mechanical strength, and color alteration. Furthermore, their utilization has been linked to the formation of micro-layers that, over time, contribute to the stone’s deterioration. As a result, their use is no longer recommended [13], and caution is warranted, as their potential drawbacks can outweigh their intended benefits. Other consolidants commonly used for stone restoration, such as alkoxysilanes that improve some mechanical properties, have shown some disadvantages, producing a layer that occludes the pores and/or the formation of an inner micro-layer within the stone pores, leaving gel residues, which may crack or break over time [19]. In fact, crack formation and detachment have been previously documented in cases where nanosilica and ethyl silicate were applied to different porous substrates [20,21]. These occurrences cast doubt on the enduring efficiency of consolidants based on ethyl silicate for the preservation of some stone types, particularly in relation to substrates rich in clay content, which exhibit fluctuations in size (expansion and contraction) due to moisture contents [22,23]. In addition to these shortcomings, some changes may also occur in the original stone color, which tends to turn yellowish or darker, especially when organic polymers are used as consolidants [14]. Although these conventional consolidants are being improved by modifying particle size and composition by creating nanoparticles that reduce or avoid side effects, they are still typically forming chemical compounds that are different from the stone substrate [19,24,25,26], or their effects are very superficial [27]. On the other hand, some authors warn that nanotechnology can generate certain risks as it can affect human and animal health, altering the environment [28]. However, nanotechnology is promising for the future as new advances, such as silica-functionalized nanolimes for the conservation of stone heritage, are being developed [29].

Currently, before restoring any building of significant historical value and in order to make the correct choice of which treatment to employ, it is necessary to test and evaluate the effectiveness of several consolidants [13,14,15,30]. Furthermore, although the suitability of the consolidants is proven in other similar cases, each type of stone is characterized by intrinsic properties that are unique, making more studies essential to check the consolidant effectiveness and behavior in each case.

This article focused on the consolidation and conservation of two historical buildings in Alicante (Spain), the Basilica of Santa Maria in Elche (hereinafter referred to as SM) (Figure 1a), and the Church of Nuestra Señora de La Asunción in Biar (hereinafter referred to as IB) (Figure 1b), through the application of a bioconsolidation treatment [31,32,33].

Both buildings were made of stone extracted from quarries located near the towns: Elche, Sierra del Ferriol at Serravallian’s age calcarenite outcrops [34,35] and Biar at Eocene’s age calcarenite outcrops [36]. Both stones are porous bioclastic calcarenite but with some differences in their composition. The stone of SM shares similarities with the lithologies found in the famous Iberian sculpture “the Lady of Elche”, which contains clays as one of its components, while that of IB is denser and contains a lower proportion of clays.

Serious stone damage was evident in both buildings showing severe alterations and extensive surface sanding caused by the granular disintegration of whole pieces. For example, Figure 1b illustrates the main entrance of Biar church, featuring a remarkable Proto-Renaissance-style façade dating back to the early sixteenth century, where significant areas of its external surface have experienced a loss of thickness exceeding 15 cm.

The bioconsolidation treatment employed here had been successfully applied in some historical buildings in Granada, like the Royal Chapel, San Jeronimo Monastery, and the Royal Hospital [37], all of them constructed using a local porous calcarenite (bioclastic limestone) that exhibited varying degrees of decay. As previously described by the authors, the evaluation process of this innovative bioconsolidation treatment, which has been closely monitored over several years, has demonstrated its excellence and effectiveness in conserving porous limestone. In fact, this novel consolidation method has proven to be a successful alternative to conventional treatments, providing an environmentally friendly strategy that harnesses bacterial carbonatogenesis (i.e., bacterially induced precipitation of calcium carbonate) to ensure the efficient preservation of historical stone. Commonly, exogenous or stone-isolated single bacterial cultures have been employed in biomineralization treatments. However, this recent and streamlined bioconsolidation approach has emerged, involving the utilization of a patented sterile nutritional solution known as M-3P [32], serving as a source of calcium and amino acids. This solution selectively activates the carbonatogenic bacteria among the stone autochthonous microbial community, thereby eliminating the need to isolate and culture such bacteria in a laboratory setting before application. The resulting bacterial calcium carbonate (calcite and/or vaterite), together with the mineralized bacterial cells and the bacterially derived exopolymeric substances (EPS) (composed of polysaccharides, proteins, DNA, and RNA [31,32]), are the main components of the biocement that consolidates the weathered stone. Moreover, according to the results obtained from those previous studies, the significant degree of consolidation was long-lasting over time without changing the stone’s color or inducing any detrimental side effects. Notably, this solution refrains from stimulating other potentially harmful microorganisms, including acid-producing, sulfate-oxidizing bacteria, nitrate-reducing bacteria, or fungi. Thus, its selectivity specifically targets the carbonatogenic bacteria while avoiding any activation of detrimental counterparts that could develop on the stone after the treatment, as shown by the cultivable and total microbiota analyses [32,37]. The patented M-3P nutritional solution has been meticulously formulated to be compatible with all materials containing calcitic substrates. Its resultant effect is the formation of biogenic calcium carbonate cement, which serves to safeguard and reinforce weathered carbonate stones. By virtue of its meticulously curated composition, encompassing amino acids, calcium, and more (see below), this sterile nutritional solution operates without any external introduction of bacteria. It activates the autochthonous carbonatogenic microbiota inherently present within the stone, thereby facilitating the precipitation of calcium carbonate crystals within the stone matrix. Notably, the meticulously curated composition of this patented nutritional solution necessitates no alteration, as it inherently adapts to all calcium carbonate or calcarenite stones. The solution only interacts with the resident bacterial communities within the stone, steering these bacteria toward the formation of calcium carbonate cement, a pivotal process that culminates in the consolidation of calcitic stone substrates. While this approach has demonstrated successful results in limestone, carved tuff stone, and gypsum plaster [21,37,38], its potential for consolidating clay-containing calcarenite stone heritage remains unexplored.

The aims of the present study are directed at assessing, on the one hand, the effectiveness of the bioconsolidation treatment on calcarenites with distinct compositions like those employed in the construction of SM and IB. On the other hand, to broaden the scope of existing research on bioconsolidation by exploring its application to other varieties of calcarenite and/or other ornamental materials [37,38,39]. In pursuit of these objectives, crucial petrophysical aspects, in addition to petrographic properties, were analyzed. Key parameters such as water vapor permeability, water absorption at low pressure, changes in the water contact angle of the stone, and increases in the drilling resistance of the treated samples were examined. This last test provides valuable insights into the treatment’s consolidation efficacy and the depth of its effects in the stone. The outcomes of our research highlight the systematic advantages of this innovative bioconsolidation method, providing a foundation for future advancements in the development of more tailored products and treatment techniques for consolidating diverse types of calcarenite stones. This study not only demonstrates the efficacy of the current approach but also paves the way for refining and expanding conservation strategies to enhance the preservation of calcarenite-based historical structures.

## 2. Materials and Methods

### 2.1. Sampling

Following the recommendations of the Spanish Institute of Cultural Heritage (IPCE) [13], block remains that were replaced during previous conservation restorations of both historical buildings served as SM and IB stone samples in this study. In the case of IB, a conservation intervention took place at the end of the 19th century, during which part of the principal façade was replaced. As a result, some of the original stone blocks were left inside the church, and a selection of these blocks were used here as IB samples. Similarly, in the case of SM, the samples used for the consolidation study were the remains of stone blocks obtained from a conservation restoration performed at the beginning of the 20th century. During this restoration, the dome stone was replaced with another one made of ceramic bricks. The remains of the original stone block were preserved and stored outdoors on the rooftop of the church.

The stone blocks from both buildings were carefully cut into pieces according to the dimensions specified in the different standards [40,41,42,43,44,45] or reference documents [46]. Thus, in the case of the SM stone, a total of 20 cubes, each measuring 4 cm × 4 cm × 4 cm in size (determined by reference norm), were obtained, along with 9 discs measuring 6.7 cm in diameter and 2.5 cm in thickness (the diameter is determined by the test glass container and the thickness by reference norm). Additionally, several irregular-shaped fragments, referred to as FG-SM (Figure 2a), were included in the study. In the case of the IB stone, 14 cubes, each measuring 4 cm × 4 cm × 4 cm in size, were prepared, in addition to 9 discs measuring 6.7 cm in diameter and 2.5 cm in thickness. As with the SM stone, several irregular-shaped fragments designated as FG-IB (Figure 2b) were also used to test the maximum amount of material available.

### 2.2. Petrographic and Petrophysical Characterization

Petrographic characterization of the treated and untreated samples was performed using:

Polarizing petrographic microscope (MOP): Zeiss Axioskop transmitted light optical microscope has been used, with a Zeiss Stemi SV 6 magnifying glass and a Photometrics Cool SNAP-CF camera. Thin sections of untreated stone have been observed. This study was carried out at the University of Alicante.

Scanning electron microscope (FESEM): Samples have been observed in two different microscopes, at the University of Granada with a Zeiss Model SMT AURIGA Field Emission Scanning Electron Microscope and at the University of Alicante with a ZEISS model Merlin VP Compact. The samples studied were small and prismatic, less than 0.7 cm × 0.7 cm × 0.7 cm. Untreated and treated stone samples have been observed.

X-ray diffraction (XRD): The Bruker D8-Advance equipment with the Göebel mirror has been used. Two pulverized grams of each type of untreated stone have been analyzed with the following conditions: Cu Kα radiation, 40 kV, 40 mA, 0.05 2Theta degrees steps, 3 s per step, and 4–60 scan angle. This analysis was carried out at the University of Alicante.

Mercury intrusion porosity (MIP): The equipment used was a POREMASTER-60 GT, QUANTACHROME INSTRUMENTS. The samples studied were small and prismatic in shape, less than 0.7 cm × 0.7 cm × 2 cm. This study was carried out at the University of Alicante.

The different standard tests (described below) were performed both before and after treatment application, with dried samples taken to constant weight. Non-destructive tests were conducted on the same samples before and after the bioconsolidation treatment to assess their effectiveness.

### 2.3. Bioconsolidation Treatment

The sterile nutritional solution M-3P [32] was applied by spraying twice a day for 7 consecutive days (Figure 2a). This frequent application was essential to maintain the stone in a saturated state throughout the treatment period. The composition of the M-3P nutritional solution includes 1% Bacto-Casitone (a hydrolyzed casein), 1% Ca(CH_3_COO)_2_·4H_2_O (total calcium: 43.44 mM), 0.2% K_2_CO_3_·1/2H_2_O (total potassium: 35.6 mM; total carbonate: 17.8 mM), and 10 mM phosphate buffer in distilled water (pH 8). Both during treatment and up to two/three days after its completion, the treated blocks remained covered with aluminum foil (stiffened with metal wires) to avoid the direct effect of sunlight and to minimize the evaporation of the M-3P solution. The treatments were conducted with strict temperature control to maintain a range of 20 to 30 °C on the treated areas. All samples were kept for at least 30 days before performing any treatment evaluation. Figure 2 shows the samples during the application of the treatment and their appearance afterward.

### 2.4. Test Descriptions for Treatment Evaluation

The most important stone property that should be improved after successful stone consolidation is the cohesion of disaggregated particles. This enhanced cohesion leads to a significant increase in the stone’s resistance to external forces and environmental deterioration. Many authors consider that the most appropriate method for the evaluation of grain cohesion, and thus of the consolidation performance, is the application of the Drilling Resistance Measurement System (DRMS, SINT Technology) [47,48,49,50] (Figure 3a). On the other hand, a successful stone consolidation should maintain or improve vapor permeability while reducing liquid water permeability through increased water static contact angle [51].

Both untreated and treated stone cubes measuring 4 cm × 4 cm × 4 cm in size were used for the evaluation of the drilling resistance of the stones. DRMS is designed by SINT Technology (Calenzano, Italy) to perform a precise ‘‘drilling resistance’’ test through the continuous measurement of the force necessary to drill a hole in the material under specific operating conditions. The test conditions were as follows: (i) a 4.8 mm diameter flat-edged diamond-tip drill bit [49]; (ii) a penetration rate set at 20 mm/min; and (iii) a rotational speed set at 200 rpm. The system is equipped with software that allows continuous recording (giving data every 0.1 mm drill) of the force, expressed in N, in relation to the drilling feed [49]. During testing, both rotational speed and penetration rate were maintained constant. Five drill holes per specimen were performed to obtain average values. This test was carried out in the Department of Physical Chemistry at the University of Cádiz.

### 2.5. Water Absorption under Low Pressure

This test was performed following the recommendations of RILEM 25 PEM test II.8 standards [44]. The test involved dropping a 1 mL water droplet onto the surface of the specimen from a distance of 3 cm. The water was stained with methylene blue to evidence the drop expansion and its absorption by the stone specimens. A stopwatch measured the time the drop took to be completely absorbed by the stone. The test was performed in the laboratory at 22 °C. This study was carried out at the University of Alicante.

### 2.6. Static Contact Angle (Ɵ) Measurement

The stone samples were subjected to this test as recommended in UNE-EN 15802 [45]. Control and treated samples were used for the treatment evaluation (in this case, only two samples of each stone type were available). For each sample, a micro-droplet of distilled water was applied with a needle onto the stone surface, generating a determined static contact angle. The image of the droplet was immediately captured for the measurement of the static contact angle (Figure 3b). The whole process was controlled on a monitor that amplified the image for a higher measurement control. This test was carried out at the Department of Physical Chemistry at the University of Cádiz.

### 2.7. Water Vapor Permeability

Following the recommendations of RILEM 25 PEM test II2 [46], this test was performed on the same samples before and after the application of the bioconsolidation treatment. To this end, three samples of 6.7 cm in diameter and about 2.5 cm in thickness were used. Each one was taken to a constant weight; the contour of the specimen was waterproofed and introduced in glass containers with approximately 80 g of completely dry silica gel. The test specimen was placed in the upper part of the container and sealed with silicone. Three test pieces were placed on a grid in a climate chamber with a layer of water provided at the bottom. To determine the amount of water vapor that was driven through the samples, measurements of the weight of the assembly (container with silica gel and the stone specimen) were carried out once a day for 30 days. The increase in weight detected was due to the water vapor absorbed by the silica gel, which had passed through the stone pores. This study was carried out at the University of Alicante.

### 2.8. Color Measurements

This test was carried out according to the UNE-EN 15886 standard [52] with the aim of measuring color changes between treated and untreated samples. Chromatic measurements were monitored before and after treatment in both buildings by using a Minolta Konica Meter (Model 2600d) spectrophotometer, including a CIE standard observer of 10°, illumination of D65, and a spectral bandwidth of 10 nm. Total color variations were reported as Δ*E* = (Δ*L**^2^ + Δ*a**^2^ + Δ*b**^2^)^½^, where Δ*L**, Δ*a**, and Δ*b** are the difference between untreated and treated stone of *L** (lightness: 0 being black and 100 being diffuse white), *a** (negative values indicate green while positive values indicate magenta), and *b** (negative values indicate blue and positive values indicate yellow). This study was carried out at the University of Alicante.

## 3. Results

### 3.1. Petrographic and Mineralogical Characterization

Data obtained from visualizing the samples using a petrographic polarizing microscope (Figure 4) and a scanning electron microscope (Figure 5) revealed that the stone of SM is a somewhat loamy biomicritic limestone. This type of limestone contains various fossils, including foraminifers, brachiopods, and echinoderms. Additionally, observations identified the presence of clays, particularly illite, as determined by XRD analysis. These clay minerals were transported from the continent and deposited along with the calcareous fossils in a shallow marine environment, subsequently subjected to cementation during the diagenesis of these calcarenite rocks. This stone composition implies a higher intrinsic predisposition to decay by extrinsic agents. Specifically, the stone is prone to phenomena such as disaggregation, flaking, and scaling of the stone surface when in contact with water. In terms of mineralogical composition, determined by XRD analysis, this stone contains 88% calcite, 2% quartz, and 10% dolomite. IB stone was shown to be a fine-grained biocalcarenite (biosparitic limestone), containing terrigenous materials with different concentrations of quartz and feldspar, oscillating between 5% and 15%. It contains fossils mainly formed by planktonic foraminifers (commonly represented by Globigerinoides), red algae, and bryozoans. According to XRD analysis, the mineralogical composition of this stone consisted of 94% calcite, 3% quartz, and 3% microcline.

Given these specific characteristics and vulnerabilities of both types of calcarenite, bioconsolidation treatments are essential for the conservation of buildings constructed with such stones. Furthermore, these stones present a high susceptibility to deterioration caused by salts, atmospheric agents, and capillary absorption of moisture [2,4,5,6,7,8,9,30], which underscores the necessity of effective conservation measures. Therefore, bioconsolidation treatments offer a promising approach to strengthen and protect the stone, thus mitigating the effects of external agents and helping to prolong the life of historical buildings and preserve their cultural and architectural significance for future generations.

SEM photomicrographs of treated stone samples showed newly formed cement of calcium carbonate in both historical buildings made with SM and IB stones (Figure 5). In both cases, based on the microstructural morphologies observed, spherulitic-shaped crystals, as well as calcified EPS and calcified bacterial cells (i.e., bacterial cells embedded or entombed within the bacterially produced calcium carbonate crystals), were distinguished within the stone substrates. The newly formed bacterial calcium carbonate cemented the stone substrate and improved its resistance to decay (see below). Moreover, as shown in Figure 5b,d, the EPS produced by the carbonatogenic bacterial activity is able to cover and bind calcarenite grains and newly formed bacterial calcium carbonate. Such an effect may contribute to the reduction of water absorption, as previously demonstrated by Le-Métayer-Levrel et al. [53]. Furthermore, the fine, newly formed biocement consolidated the stone without blocking or plugging the stone pores, thus allowing gas exchange with the outside. These results were consistent with previous research involving the same bioconsolidation treatment on historical buildings in Granada, i.e., San Jeronimo Monastery, Royal Hospital, and Royal Chapel. Despite the different types of stone used in those buildings (calcarenite without clays) and a distinct mineralogical composition (over 95% calcite and less than 5% quartz) from that of SM and IB stones, the results demonstrated noticeable and lasting strengthening effects associated with the production of bacterial calcium carbonate cement [31,33,37]. In those projects, in all cases, the bioconsolidation treatment proved to be effective in the medium- and long-term, significantly enhancing the structural integrity of the stone and providing enduring conservation benefits [37,54]. These consistent outcomes across various stone types and mineralogical compositions, including those containing some clays in their composition, such as those of SM and IB, highlight the reliability and potential of the bioconsolidation approach as a valuable and lasting conservation strategy for historical buildings constructed from calcarenite materials.

### 3.2. Petrophysical Characterization

Among the petrophysical characterization study, the most relevant data belongs to porosity and capillary absorption. Results obtained from the Mercury Intrusion Porosity (MIP) test (Figure 6) indicated that in both cases, the volume of porosity before treatment was very high, resulting in a mean value of 36.6% in the case of SM and of 30.2% in that of IB. Capillary absorption coefficients [43] of SM- and IB-untreated samples were 26 g/m^2^s^0.5^ and 30 g/m^2^s^0.5^, respectively.

Comparing the MIP data of treated and untreated samples, a reduction of hardly 1% porosity was observed in SM samples, while no detectable reduction was observed in the case of IB samples, maintaining the average stone total porosity. MIP curves moved slightly toward lower diameters but remained in similar ranges so that the behavior against capillary phenomena or salt crystallization, or freeze–thaw cycles were not observed to be affected [55]. These results further confirmed that the newly formed abundant bacterial CaCO_3_ cement consolidated the stone without pore blocking, as was previously shown by SEM analyses. The same effects had been previously demonstrated after the application of this bioconsolidation treatment on other historical buildings [37,54]. The high capillary absorption coefficient indicated that when applying M-3P treatment, the liquid penetrated easily into the stone samples [55,56], being faster in the case of IB.

### 3.3. Drilling Resistance Measurement System “DRMS”

Drilling resistance curves of SM samples and IB samples are presented in Figure 6, where the average values of five drill measurements are plotted in Newtons in accordance with the depth of drilling in mm. Blue curves showed the results before the treatment and the red ones after the bioconsolidation treatment. In both cases, after the bioconsolidation treatment, a clear increase in drilling resistance was observed, especially in the first few millimeters, where it doubles in comparison with the untreated samples. In the case of SM samples, significant strengthening is observed up to a depth of 4 mm (Figure 7a). In the case of IB samples, the strengthening effect on the treated stones was very high and extended up to a depth of 14 mm (Figure 7b). This greater depth of consolidation might probably be related to the high capillary absorption coefficient and to the different mineralogy of each type of stone [57]. Note that the best consolidation effect in terms of drilling resistance was observed following the bioconsolidation treatment applied in the case of IB. In addition, the resistance of both stones before and after bioconsolidation treatment was more noticeable in IB than in SM. This is probably due to the lower porosity of IB stone.

These results further corroborate that the strengthening achieved is crucial for the successful consolidation of the stone. Here, once more, the effectiveness of the bioconsolidation treatment applied to calcarenite samples was confirmed, also being aligned with the previous studies [32,37,38,39].

### 3.4. Water Vapor Permeability

The results of water vapor permeability are shown in Figure 8. The values have been determined as vapor permeability in accordance with the weight. The curves showed a slight reduction of permeability in both cases. Quantification of the mean weight increase and calculation of the coefficient of water vapor permeability were performed for each case. After 30 days, the SM-untreated sample allowed the transit of 6 g of water vapor, while the treated one allowed 5.85 g, so the permeability was only reduced by 2.5%. The coefficient of water vapor conductivity of the SM-untreated sample was δ = 6.56 × 10^−4^ (g/m^2^), and that of the treated sample was δ = 6.36 × 10^−4^ (g/m^2^s). Concerning IB samples, after 30 days, the untreated sample allowed the transit of 5.76 g of water vapor, while the treated one allowed 5.53 g, so the permeability was reduced by about 4 %. The coefficient of water vapor conductivity of the untreated sample was δ = 6.30 × 10^−4^ (g/m^2^s), and that of the treated sample was δ = 6.05 × 10^−4^ (g/m^2^s).

These data indicated and confirmed the good behavior of SM and IB stones after the bioconsolidation treatment tested here. The water vapor permeability in the studied stones was reduced much below the limits recommended by experts, who fixed it at 50% [49]. These results are in alignment with those obtained by the MIP measurements referred to above.

### 3.5. Static Contact Angle Test

This test helps describe how a liquid droplet interacts with a solid surface and refers to the angle formed between the tangent line of the liquid droplet’s surface where it meets the solid surface and the stone surface itself. This angle is measured within the liquid droplet at the point of contact. The results obtained by the static contact angle measurements are shown in Table 1. These measurements showed a high increase in the static contact angle values after treatment in both types of stone, being more significant in the case of SM. It seems that the surfaces of treated samples become somehow “waterproofed” as a consequence of the bioconsolidation treatment, i.e., it is more difficult for water to adhere to the stone surface, thus improving this feature in comparison to untreated stones. According to Jroundi et al. [31], this would be achieved by inducing the formation of an abundant amount of exceptionally strong CaCO_3_ cement that incorporated bacterially derived organics. This may also be due to modification of the topography of the stone surface [58], with the shape of the consolidating bacteria, which are mainly rod-shaped and generate an increase in the static contact angle.

### 3.6. Water Absorption Rate at Low Pressure

The results of this test are presented in Table 2, showing that in both cases, an increase in the absorption time of the water drop occurred, being much more noticeable in the case of IB samples. Because of the porosity of the stone, results are very heterogeneous, depending on the exact spot where the drop has fallen, which is a common occurrence in this type of stone [47]. During this test, all treated samples showed a repellent effect against water; the droplets expanded less and showed more contained edges, as shown in Figure 9, in which two droplets were deposited simultaneously in areas A (untreated) and B (treated) of an SM fragment. In all consolidation treatments, considerably lower values of water absorption rate were observed. These results can be explained by the fact that new consolidation materials were precipitated in both samples—results previously shown in this work obtained by the SEM analyses discussed above.

### 3.7. Color Measurements

Table 3 and Figure 10 present results obtained on stone color measurements before and after bioconsolidation treatments in SM and IB samples. The curves of each type of stone (Figure 10) demonstrated similarity before and after the application of the treatment, meaning that no significant differences in color were observed. This result is particularly noteworthy, as one of the primary concerns regarding in situ treatment application is the potential for visible color changes in the treated stone. However, our study’s results showed that the bioconsolidation treatment produced no significant color changes in both SM and IB stones. The Δ*E* values after treatment were 1.08 and 1.90 in SM and IB, respectively. These values are below 2, which is considered an acceptable threshold value for any stone treatment, and they are regarded as extremely low according to the ISO 12647 standard [59,60,61], as demonstrated by numerous previous studies [61,62]. These findings provide reassurance that the in situ application of the bioconsolidation treatment does not produce any undesirable and visually apparent alterations to the original appearance of calcarenite stones. In addition, the results demonstrated similar color outcomes to those previously obtained in many historical buildings, such as San Jeronimo Monastery, Royal Hospital, and Royal Chapel in Granada, Spain [21,37], all of which are constructed from a porous bioclastic calcarenite stone, different in composition from the stones studied here.

The consistent and unaltered color results across different types of calcarenite stones in various historical buildings underscore the effectiveness and reliability of the bioconsolidation treatment as a safe and aesthetically sensitive conservation approach for preserving such important cultural heritage.

## 4. Discussion

The study presented here encompasses a comprehensive exploration of various aspects related to the preservation and conservation of historical calcarenite stone structures of different compositions from those previously studied. This study engages in a comprehensive exploration of a promising conservation solution: bioconsolidation treatment. The ongoing results highlight the significance of the innovative approach, its application, and the implications for the broader field of heritage preservation.

Historical buildings, often constructed with calcarenite stones, stand as a testament to human history and culture. However, they are confronted with the relentless impact of time, weathering agents, pollution, and salt crystallization, particularly in humid environments. As a result, the gradual increase in material porosity leads to a cascade of issues, including decohesion, mineral constituent disintegration, and, ultimately, structural decay. Preserving these types of historical architectural structures commonly presents a multifaceted challenge.

Therefore, central to this study is the concept of bioconsolidation treatment, a novel approach that harnesses the potential of indigenous carbonatogenic bacteria within the stone. The patented nutritional solution, M-3P, acts as a catalyst, prompting these bacteria to precipitate calcium carbonate crystals, thereby fortifying the stone’s structural integrity. This strategy proves effective in consolidating weathered areas, as substantiated by the Drilling Resistance Measurement System (DRMS) results. The efficacy of this bioconsolidation treatment was also evidenced through various analyses, including petrographic and petrophysical characterization, water vapor permeability, and color analysis. Results indicated enhanced stone cohesion, reduced water permeability, and improved hydrophobicity without compromising vapor permeability or altering the stones’ color. These outcomes were consistent with previous studies conducted on similar historical structures, further validating the method’s reliability and potential for broader application.

Additionally, through FESEM analysis, the presence of calcified bacterial cells and the detection of bacterial calcium carbonate covered by calcified EPS were evidenced. The method’s effectiveness in consolidating calcitic substrates is further corroborated, and concerns about potential color changes are alleviated, as demonstrated by minimal Δ*E* values. This work highlights the significance of long-term effectiveness, particularly for clay-containing calcarenite substrates. By comparing and contrasting this innovative approach with traditional consolidants, the advancement in conservation strategies ensures the longevity and integrity of our cultural heritage. For instance, comparative analysis with conventional consolidants, such as nanolimes and ethyl silicate, offers insights into the advantages and limitations of each method. While traditional consolidants have demonstrated efficacy, they also raise concerns regarding color changes and potential side effects on the stone, resulting in the formation of cracks and fissures within the stones altering thus the effectiveness of this treatment. In contrast, bioconsolidation treatment distinguishes itself with its non-destructive nature and minimal impact on stone aesthetics, thereby mitigating apprehensions associated with visual alterations.

Furthermore, the consistent success of the treatment across various historical buildings underscores its broader potential. By comparing outcomes to prior studies, including the San Jeronimo Monastery, Royal Hospital, and Royal Chapel, this study establishes the method’s reliability across diverse stone compositions. The implications are far-reaching, potentially revolutionizing the conservation landscape by offering a sustainable and effective alternative for preserving calcarenite stones in an array of architectural and historical contexts.

Finally, this innovative approach not only addresses the tangible issues of deterioration but also offers a glimpse into the future of heritage preservation by expanding to the consolidation of different types of historical calcarenite stone substrates. The crossroads of science, conservation, and cultural heritage converge, presenting a formidable opportunity to safeguard our shared history for generations to come.

## 5. Conclusions

The comprehensive study and analysis of the results, along with visual inspections of the samples before and after the bioconsolidation treatment, confirm the effectiveness of the bacterial carbonatogenesis achieved by the application of the nutritional solution M-3P in strengthening the tested stone. In the case of SM samples, an increase in the drilling resistance was reached in the first millimeters of the stone surface. For the IB samples, the resistance notably improved up to 14 mm depth, doubling its surface resistance, as indicated by the DRMS results. This difference in consolidation depth is attributed to the faster penetration of the M-3P nutritional solution in the case of IB samples, possibly due to the combined effect of the spraying method used for treatment application and the low porosity of this particular sample. Moreover, in both cases, the static contact angle increased after the treatment exceeded double its original values. This indicates that water drops have greater difficulty adhering to the consolidated surface, making the stone more hydrophobic. The increased absorption time of water drops further demonstrates the treatment’s effectiveness in improving the stone’s performance against disturbances generated by humidity.

Regarding water vapor permeability, the bioconsolidation treatment to produce the new cement was successful in consolidating the stone without blocking its pores. As a result, there was no significant difference in water vapor permeability between the treated and untreated samples. This is a positive outcome as many conventional consolidation treatments often reduce water vapor permeability [47], which is a critical property for further assessing the resistance to evaporation of humidity contained within the stone. SEM observations of the samples revealed the presence of newly formed crystals of calcium carbonate, along with calcified bacterial cells and calcified EPS. These components all contributed to the consolidation of the stone substrate and improved its resistance. In addition, no color changes were observed after the treatment, affirming the successful preservation of the stone’s original appearance. Overall, these results demonstrated the excellence and effectiveness of this bioconsolidation treatment on calcarenites with different compositions since it not only enhances the stone’s resistance but also maintains its water vapor permeability and original color. The present successful application of the bioconsolidation treatment extends the existing body of knowledge and strengthens the evidence of its efficacy on different historical buildings, further validating its potential as a promising conservation approach for preserving calcarenite stones in various architectural and historical contexts.

## Figures and Tables

**Figure 1 materials-16-06043-f001:**
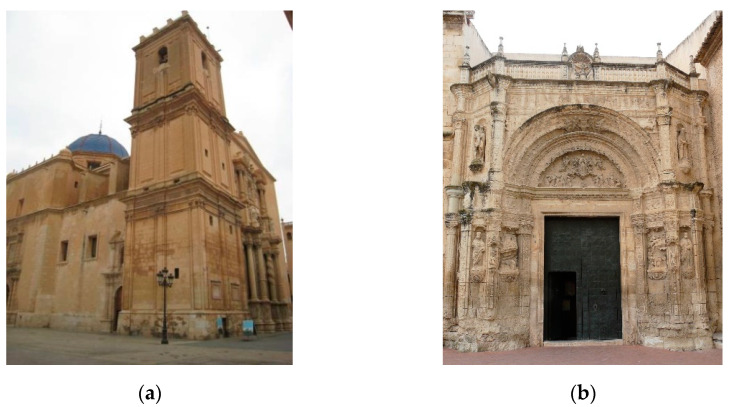
Historical buildings built with calcarenite stone; (**a**) Basílica Santa Maria (SM) in Elche (Serravallian stone); (**b**) Church of Nuestra Señora de La Asunción in Biar (IB) (Eocene stone).

**Figure 2 materials-16-06043-f002:**
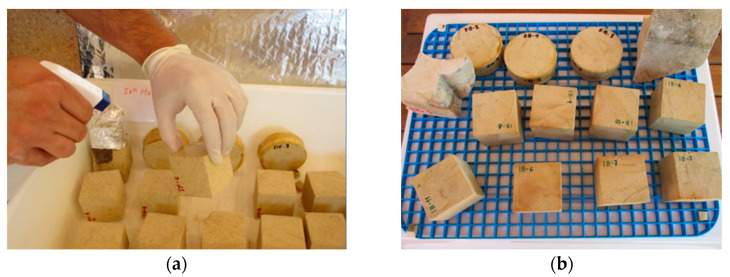
Appearance of the SM and IB samples used for bioconsolidation treatment; (**a**) Application of the treatment by spraying M-3P on the samples; (**b**) Disposition of the samples during the time the treatment lasted.

**Figure 3 materials-16-06043-f003:**
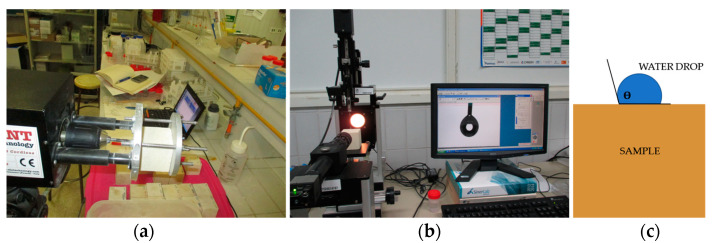
Different aspects of the tests; (**a**) DRMS; (**b**) Static contact angle test; (**c**) Contact angle test scheme. Essays carried out by the authors at the University of Cádiz.

**Figure 4 materials-16-06043-f004:**
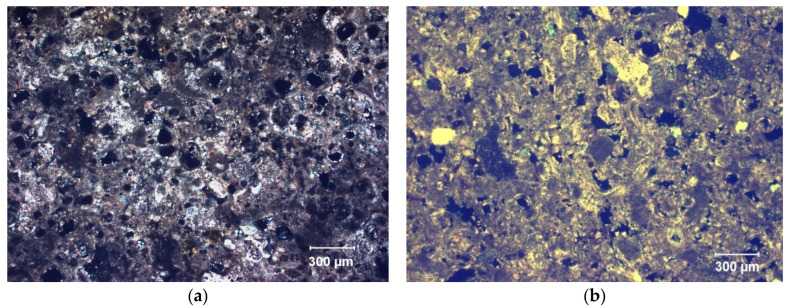
Microphotographs of thin sections (petrographic polarizing microscope with crossed Nicols); (**a**) SM sample, biomicritic limestone with foraminifers; (**b**) IB simple, biosparitic limestone with fossils (bryozoans and red algae). Both samples show irregular and heterogeneous porosity (pores are black).

**Figure 5 materials-16-06043-f005:**
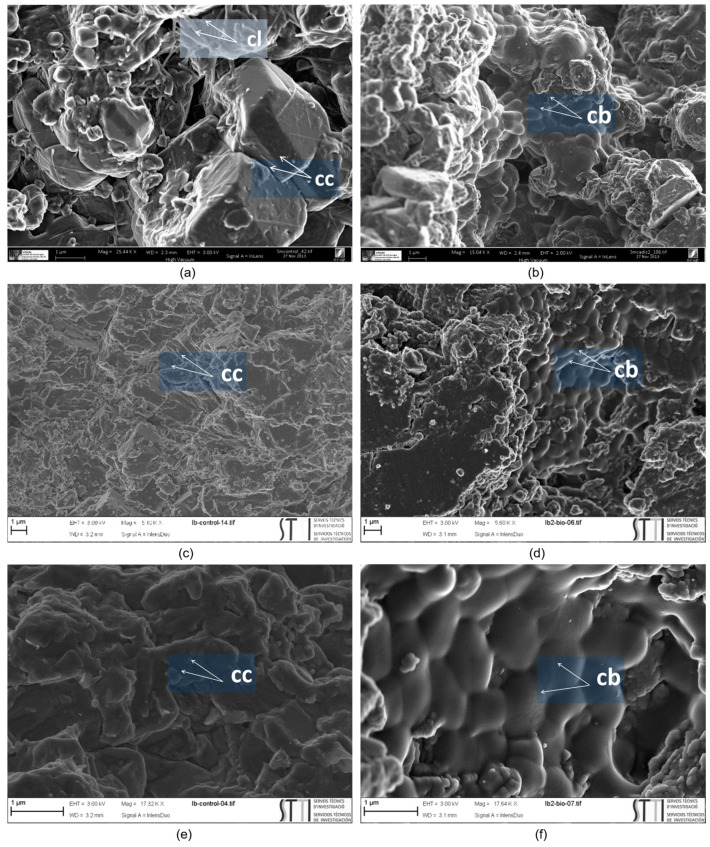
SEM photomicrographs of the samples; (**a**) SM-untreated stone; (**b**) SM-treated stone showing biomineralization; (**c**) IB-untreated stone; (**d**) IB-treated stone showing biomineralization; (**e**) Detail of IB-untreated stone; (**f**) Detail of IB-treated stone showing biomineralization and a cover of calcified bacteria; **cb** = calcified bacteria, **cc** = calcium carbonate crystals, **cl** = clays.

**Figure 6 materials-16-06043-f006:**
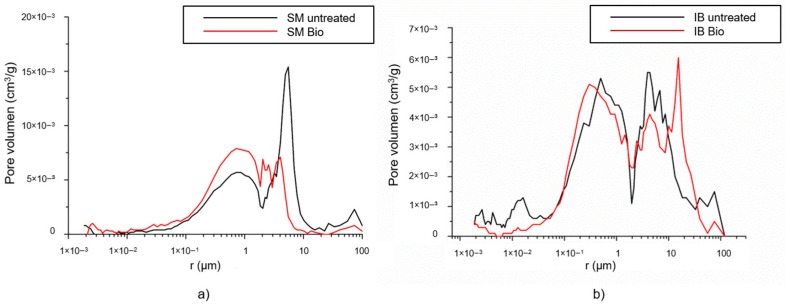
Comparison of mercury intrusion porosimetry curves of treated and untreated samples; (**a**) SM samples; (**b**) IB samples. Black lines correspond to untreated samples and red lines to treated samples.

**Figure 7 materials-16-06043-f007:**
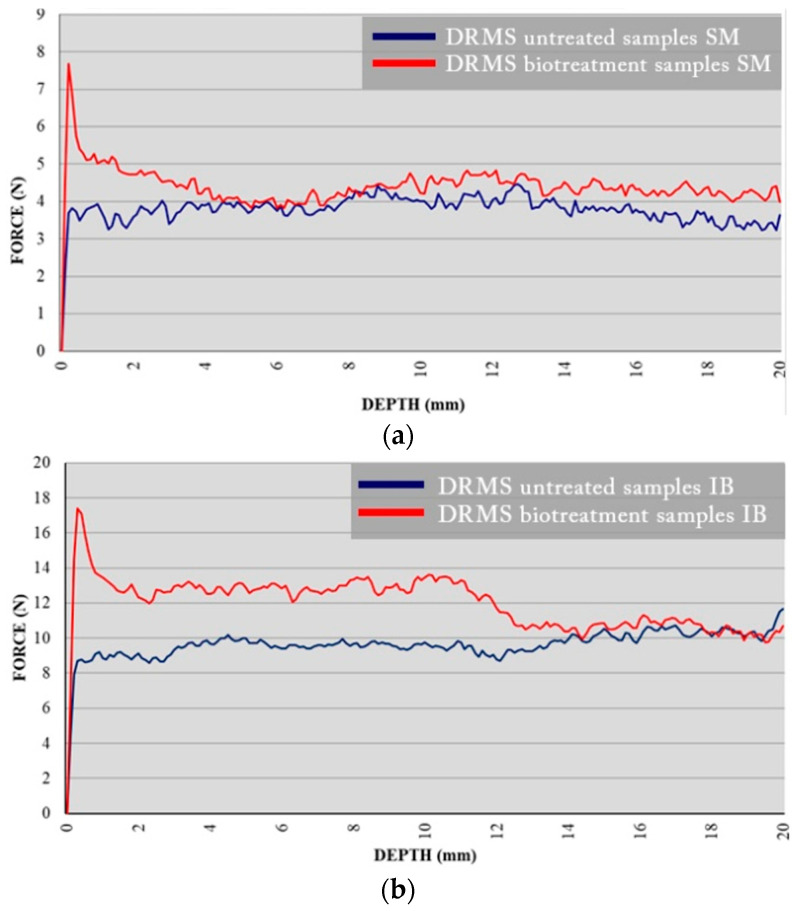
Results of DRMS test performed show the resistance in N that opposes the material to be drilled as it advances inside the sample; (**a**) SM specimens; (**b**) IB specimens. The blue and red curves correspond to the untreated and bacterially treated samples, respectively.

**Figure 8 materials-16-06043-f008:**
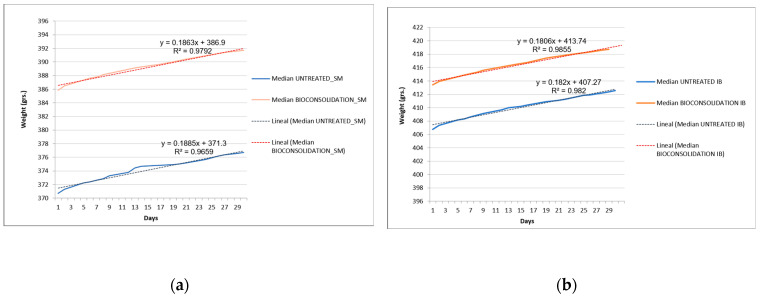
Permeability to water vapor curves; (**a**) SM-treated and untreated samples; (**b**) IB-treated and untreated samples. The blue and red curves correspond to the untreated and bacterially treated samples, respectively. The slopes of the curves are included. The initial weight in each case is different due to the weight of the containers. In both cases, the slope of the curve is slightly reduced with the biotreatments.

**Figure 9 materials-16-06043-f009:**
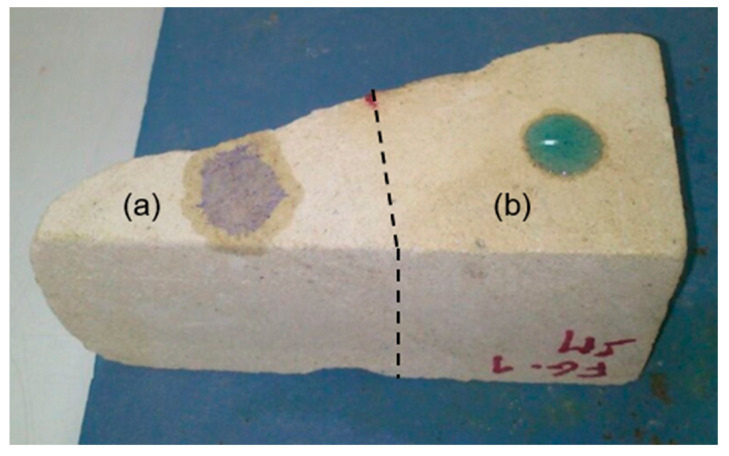
Droplet absorption test on the SM-FG1 stone sample; (**a**) Untreated part; (**b**) Treated part.

**Figure 10 materials-16-06043-f010:**
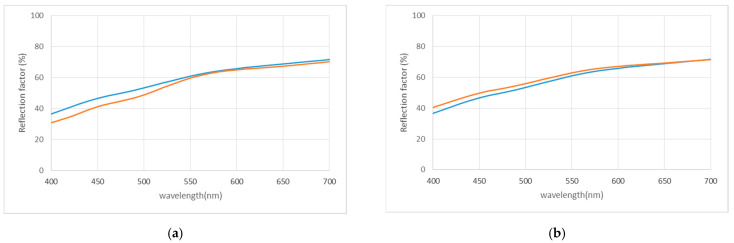
Visible reflectance spectra of treated and untreated samples; (**a**) SM; (**b**) IB. The blue and red curves correspond to the untreated and bacterially treated samples, respectively.

**Table 1 materials-16-06043-t001:** Static contact angle.

	Untreated Ɵ_s_	Treated Ɵ_s_
Average SM	25.71 ± 3.60	67.80 ± 4.55
Average IB	30.75 ± 3.59	48.85 ± 2.06

**Table 2 materials-16-06043-t002:** Water absorption rate at low pressure.

	Untreated	Treated
Average SM	35 s	1 min 40 s
Average IB	45 s	3 min 20 s

**Table 3 materials-16-06043-t003:** Spectrophotometric color measurements of the SM and IB samples before and after bioconservation treatment.

	*a**	*b**	*L**
Average SM-untreated	2.74	18.38	79.83
Average SM-treated	2.43	18.20	80.84
Average IB-untreated	2.07	13.82	81.96
Average IB-treated	1.62	12.26	82.95

## Data Availability

All data used in this research are included in the main text, tables and figures.

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
