# Peer review of "Bioconsolidation of Damaged Construction Calcarenites and Evaluation of the Improvement in Their Petrophysical and Mechanical Properties"

_materials, 2023, doi:10.3390/ma16176043_

Round 1

Reviewer 1 Report

The paper deals with an important issue - how to tackle the wheathering of stone - and describes a promissing way of conservation treatment: to use bacteria to produce the consolidant inside the object. However, the methodology was already reported elsewhere, therefore the novelty was ranked only as 'average'. The authors present two new case studies which will be, nevertheless, of high interst for the readers. 

The paper is well written and methods as well as results are presented clearly. In some cases, one or two words more of explanations would be helpful and should be placed in the introduction. The EPS (Extracellular Polymers) remain somewhat unclear. Why is the M-3P solution selective? 

I was also wondering whether you use the same type of nutrition solution for all types of calcium carbonate or calcanerite stones, resp., or if it works only for one type, or if is necessary to modify the solution in order to adapt it to a specific type of stone?  It would be very helpful if the aurhors could insert some sentences into the paper commenting to those question.

There is a typo in reference 11 : heritage instead of herigate, I guess.

Author Response

Dear Editor

Please find enclosed the revised version of our manuscript: “Bioconsolidation of damaged construction calcarenites and evaluation of the improvement in their petrophysical and mechanical properties” by Yolanda Spairani-Berrio, Jose A. Huesca-Tortosa, Carlos Rodriguez-Navarro, María Teresa Gonzalez-Muñoz, Fadwa Jroundi, submitted for publication in the Journal Materials.

We have revised the manuscript addressing all the points raised by the referees. In particular, the following has been done (the referee's comments are indicated in bold followed by our answer):

 The paper deals with an important issue - how to tackle the wheathering of stone - and describes a promissing way of conservation treatment: to use bacteria to produce the consolidant inside the object. However, the methodology was already reported elsewhere, therefore the novelty was ranked only as 'average'. The authors present two new case studies which will be, nevertheless, of high interst for the readers. 

The paper is well written and methods as well as results are presented clearly. In some cases, one or two words more of explanations would be helpful and should be placed in the introduction.

The introduction section was modified to include more clarification.

The EPS (Extracellular Polymers) remain somewhat unclear.

The text has been modified in pg 3 lines 126-128.

Why is the M-3P solution selective? 

The M-3P nutritional solution exhibits selectivity by exclusively activating the autochthonous carbonatogenic microbiota inherent in the stone. This microbiota possesses the capability to induce the formation of calcium carbonate crystals within the stone structure. Notably, this solution refrains from stimulating other potentially harmful microorganisms, including acid-producing, sulfate-oxidizing bacteria, nitrate-reducing bacteria, or fungi. Thus, its selectivity specifically targets the carbonatogenic bacteria while avoiding any activation of detrimental counterparts.

The text has been modified to include this observation in pg 3 lines 132-135.

I was also wondering whether you use the same type of nutrition solution for all types of calcium carbonate or calcanerite stones, resp., or if it works only for one type, or if is necessary to modify the solution in order to adapt it to a specific type of stone? It would be very helpful if the authors could insert some sentences into the paper commenting to those question.

The patented M-3P nutritional solution has been meticulously formulated to be compatible with all materials containing calcitic substrates. Its resultant effect is the formation of calcium carbonate cement, which serves to safeguard and reinforce weathered carbonate stones. By virtue of its meticulously curated composition, including amino acids, calcium, and more, this sterile nutritional solution operates without any external introduction of bacteria. It activates the autochthonous carbonatogenic microbiota inherently present within the stone, thereby facilitating the precipitation of calcium carbonate crystals within the stone matrix. Notably, the composition of this patented nutritional solution necessitates no alteration, as it inherently adapts to all calcium carbonate or calcarenite stones. The solution interacts with the resident bacterial communities within the stone, steering these bacteria toward the formation of calcium carbonate cement, a pivotal process that culminates in the consolidation of calcitic substrates.

The text has been changed to include this question in pg 3 lines 137-151.

There is a typo in reference 11 : heritage instead of herigate, I guess.

The error has been corrected in the revised References Section.

We have added a figure (figure 3 pg.7) to illustrate the DRMS and contact angle tests.

Please do not hesitate to contact me as the corresponding author to discuss any further questions you may have about our on-going research, our findings or this manuscript.

Best regards,

The authors.

Reviewer 2 Report

This is an accurate study that provides some extra assessment on a promising approach to achieve bioconsolidation of stone. While the study is relevant, there are some issues that should be addressed:

The novelty of the study is only delivered at the end of the Intro, in one short paragraph. It should be better detailed, e.g. saying more extensively which types of stone characteristics were selected to cover a range of possibilities in applications.

Also: there are other studies in the literature that used bacteria to produce calcium carbonate on stone. The difference/novelty of the present study as compared to existing literature should be stressed better.

Then, the intro seems a bit rushed when it comes to putting the proposed research in the context of past and current consolidation methods. For instance, consolidants based on nanolimes, either straight or added with nanosilica or fumed silica, have been widely reported on stone, and even demonstrated/reviewed by some of the paper’s authors. These methods can be quite effective and sustainable, despite some limitations, thus their essential aspects should be concisely evaluated to see where bioconsolidation might be considered as an alternative/complementary approach.

Finally, the literature cited by the authors on traditional consolidants is mostly national and not in English, thus could be harder to find/read to the broad scientific community. I suggest replacing them as much as possible with references in English from peer reviewed international journals.

Another crucial point is that the performances of the new method should be compared at least with one-two benchmarks, e.g. other bacterial treatments or nanolimes/silica, if not experimentally at least comparing some of the observables with other studies in the literature. This should be addressed in the discussion and conclusions.

Fig. 4: what is meant, exactly, by “calcified bacteria”? How can bacterial cells be distinguished from calcified ones simply by their shape and morphological aspect under the SEM? Maybe a SEM-EDX map showing the presence of Ca over the cells would help clarify.

Fig. 6-7 might be made more readable.

Author Response

Dear Editor

Please find enclosed the revised version of our manuscript: “Bioconsolidation of damaged construction calcarenites and evaluation of the improvement in their petrophysical and mechanical properties” by Yolanda Spairani-Berrio, Jose A. Huesca-Tortosa, Carlos Rodriguez-Navarro, María Teresa Gonzalez-Muñoz, Fadwa Jroundi, submitted for publication in the Journal Materials.

We have revised the manuscript addressing all the points raised by the referees. In particular, the following has been done (the referee's comments are indicated in bold followed by our answer):

 This is an accurate study that provides some extra assessment on a promising approach to achieve bioconsolidation of stone. While the study is relevant, there are some issues that should be addressed:

We thank the reviewer for their insightful and constructive feedback, which undoubtedly enhances the caliber of our manuscript. Additionally, we appreciate the diligent effort taken by the reviewer in meticulously reviewing and revising this manuscript.

The novelty of the study is only delivered at the end of the Intro, in one short paragraph. It should be better detailed, e.g. saying more extensively which types of stone characteristics were selected to cover a range of possibilities in applications.

The information has been added in the revised text pg 3 lines 157-163.

Also: there are other studies in the literature that used bacteria to produce calcium carbonate on stone. The difference/novelty of the present study as compared to existing literature should be stressed better.

We thank the reviewer for this suggestion. The novelty and difference of the present study are stressed better in the revised version of the text pg 3 lines 103-134.

Then, the intro seems a bit rushed when it comes to putting the proposed research in the context of past and current consolidation methods. For instance, consolidants based on nanolimes, either straight or added with nanosilica or fumed silica, have been widely reported on stone, and even demonstrated/reviewed by some of the paper’s authors. These methods can be quite effective and sustainable, despite some limitations, thus their essential aspects should be concisely evaluated to see where bioconsolidation might be considered as an alternative/complementary approach.

The text has been modified according to the reviewer suggestion in pg 2 lines 68-83.

Finally, the literature cited by the authors on traditional consolidants is mostly national and not in English, thus could be harder to find/read to the broad scientific community. I suggest replacing them as much as possible with references in English from peer reviewed international journals.

The references have been modified to include this suggestion from the reviewer.

Another crucial point is that the performances of the new method should be compared at least with one-two benchmarks, e.g. other bacterial treatments or nanolimes/silica, if not experimentally at least comparing some of the observables with other studies in the literature. This should be addressed in the discussion and conclusions.

Discussion section was included as suggested by another referee and we include there this suggestion in pg. 14 lines 523-537.

Fig. 4: what is meant, exactly, by “calcified bacteria”?

Calcified bacteria mean bacterial cells embedded or entombed within the bacterially-produced calcium carbonate crystals.

A clarification has been included in the revised text pg. 8 lines 333-335.

How can bacterial cells be distinguished from calcified ones simply by their shape and morphological aspect under the SEM? Maybe a SEM-EDX map showing the presence of Ca over the cells would help clarify.

Distinguishing calcified bacterial cells becomes possible due to the presence of a delicate layer of precipitated calcium carbonate enveloping either the entirety or a portion of the cells, alongside calcified EPS. The bacterial surfaces play an important role in such a mineral formation by serving as heterogeneous nucleation sites (Fortin 1997; Rodriguez-Navarro et al. 2003). These results align with previous studies that have employed FESEM analysis to detect the emergence of bacterial calcium carbonate enveloped by EPS (Elert et al., 2021; Jroundi et al., 2017; among others). This unequivocally substantiates heightened carbonatogenic bacterial activity, serving as compelling evidence for the efficacy of the in-situ and laboratory biotreatment. However, conducting EDX analysis on such samples presents challenges, particularly in discerning the origin of calcium peaks. Such peaks could potentially denote calcium from the substrate or the newly formed calcium carbonate, rendering the analysis less suitable for this context. 

We have added a figure (figure 3 pg.7) to illustrate the DRMS and contact angle tests.

References

  • Fortin D (1997) Surface-mediated mineral development. Rev Mineral 35:162–180.

- Rodriguez-Navarro C, Rodriguez-Gallego M, Chekroun KB, Gonzalez-Muñoz MT (2003) Conservation of ornamental stone by Myxococcus xanthus-induced carbonate biomineralization. Appl Environ Microbiol 69:2182–2193

- K. Elert, E. Ruiz-Agudo, F. Jroundi, M.T. Gonzalez-Muñoz, B.W. Fash, W.L. Fash, N. Valentine, A. de Tagle, C. Rodriguez-Navarro Degradation of ancient Maya carved tuff stone at Copan and its bacterial bioconservation, npj Mater Degra, 5 (2021), pp. 1-12

- Jroundi, F., Schiro, M., Ruiz-Agudo, E., Elert, K., Martín-Sánchez, I., González-Muñoz, M. T., et al. (2017). Protection and consolidation of stone heritage by self-inoculation with indigenous carbonatogenic bacterial communities. Nat. Commun. 8:279. doi: 10.1038/s41467-017-00372-3

Fig. 6-7 might be made more readable.

Please do not hesitate to contact me as the corresponding author to discuss any further questions you may have about our on-going research, our findings or this manuscript.

Best regards,

The authors.

Reviewer 3 Report

Dear Authors,

I find your manuscript a useful and valuable contribution that fits well to the scope of the journal Materials (MDPI). The manuscript is well-written both in English and scientific argument perspectives. The research design is easy to follow and set in a logical way. In the Method section you could add some technical information about the Instruments and their technical parameters as well as where they are housed. The manuscript has some minor issues I marked in an annotated PDF provided. Probably the biggest shortfall of the manuscript is the lack of proper Discussion. A page comparative summary of comparing the method presented here with other techniques used elsewhere, something that sum the pros and cons up certainly would have been and would be some good addition. If you provide such section, the manuscript would be more relevant other researches and globally more visible. All these could be obtained through a minor revision.

Best regards.

English is fine

Author Response

Dear Editor

Please find enclosed the revised version of our manuscript: “Bioconsolidation of damaged construction calcarenites and evaluation of the improvement in their petrophysical and mechanical properties” by Yolanda Spairani-Berrio, Jose A. Huesca-Tortosa, Carlos Rodriguez-Navarro, María Teresa Gonzalez-Muñoz, Fadwa Jroundi, submitted for publication in the Journal Materials.

We have revised the manuscript addressing all the points raised by the referees. In particular, the following has been done (the referee's comments are indicated in bold followed by our answer):

 I find your manuscript a useful and valuable contribution that fits well to the scope of the journal Materials (MDPI). The manuscript is well-written both in English and scientific argument perspectives. The research design is easy to follow and set in a logical way.

We thank the reviewer for the positive feedbacks on our manuscript.

In the Method section you could add some technical information about the Instruments and their technical parameters as well as where they are housed.

The changes have been done. We have included a new figure (figure 3 pg.7) to illustrate the DRMS and contact angle tests.

The manuscript has some minor issues I marked in an annotated PDF provided.

The revised text has been modified to include all suggestions of the reviewer annotated in the PDF.

Probably the biggest shortfall of the manuscript is the lack of proper Discussion. A page comparative summary of comparing the method presented here with other techniques used elsewhere, something that sum the pros and cons up certainly would have been and would be some good addition. If you provide such section, the manuscript would be more relevant other researches and globally more visible. All these could be obtained through a minor revision.

We are thankful to this reviewer for this important suggestion. A Discussion Section is added in the revised version of the text as suggested.

Please do not hesitate to contact me as the corresponding author to discuss any further questions you may have about our on-going research, our findings or this manuscript.

Best regards,

The authors.

Round 2

Reviewer 2 Report

the Authors addressed the reviewers' criticisms and improved the quality of the manuscript